# Estrogen Actions in Placental Vascular Morphogenesis and Spiral Artery Remodeling: A Comparative View between Humans and Mice

**DOI:** 10.3390/cells12040620

**Published:** 2023-02-14

**Authors:** Mariam Rusidzé, Adrien Gargaros, Chanaëlle Fébrissy, Charlotte Dubucs, Ariane Weyl, Jessie Ousselin, Jacqueline Aziza, Jean-François Arnal, Françoise Lenfant

**Affiliations:** 1Institute of Metabolic and Cardiovascular Diseases (I2MC), INSERM U1297, University of Toulouse III-Paul Sabatier (UPS), CHU, 31432 Toulouse, France; 2Department of Pathology, Cancer University Institute of Toulouse Oncopole-IUCT, 31100 Toulouse, France

**Keywords:** placentation, trophoblasts, spiral artery remodeling, angiogenesis, hemodynamics, estrogen receptor

## Abstract

Estrogens, mainly 17β-estradiol (E2), play a critical role in reproductive organogenesis, ovulation, and fertility via estrogen receptors. E2 is also a well-known regulator of utero-placental vascular development and blood-flow dynamics throughout gestation. Mouse and human placentas possess strikingly different morphological configurations that confer important reproductive advantages. However, the functional interplay between fetal and maternal vasculature remains similar in both species. In this review, we briefly describe the structural and functional characteristics, as well as the development, of mouse and human placentas. In addition, we summarize the current knowledge regarding estrogen actions during utero-placental vascular morphogenesis, which includes uterine angiogenesis, the control of trophoblast behavior, spiral artery remodeling, and hemodynamic adaptation throughout pregnancy, in both mice and humans. Finally, the estrogens that are present in abnormal placentation are also mentioned. Overall, this review highlights the importance of the actions of estrogens in the physiology and pathophysiology of placental vascular development.

## 1. Comparative Placental Physiology and the Placental Vascular System between Human and Mouse Placenta

The placenta is an ephemeral transient organ, which ensures the development of offspring. However, placentas also exhibit strikingly variable morphologies between mammalian species that may possess significant reproductive benefits [1].

The morphology and cellular composition of the placenta are remarkably different between humans and mice (Figure 1). However, they share several structural and functional similarities, including the discoid shape. Furthermore, in humans as in mice, the maternal component of the placenta is represented by the decidua basalis (Db). Indeed, the decidua is a structurally modified uterine endometrial stroma, which is the site of implantation for the embryo when it is under the influence of progesterone. Moreover, it also plays a major role in the development of the placenta and forms a physical and biochemical barrier between the mother and the embryo. However, the timing and initiation of endometrial decidualization differ between these two species. In mice, decidualization is induced by embryo implantation, whereas in humans this process is already visible before conception, thereby indicating that the maternal tissue is ready for implantation.

In addition to the maternal decidua, the murine definitive placenta (i.e., the one present on embryonic day 10.5–11.5 [2,3]) contains two histologically and functionally distinct layers of fetal origin, which are the junctional zone (JZ) and the labyrinth (L) (Figure 1A,B) **[4]**.

The junctional zone (JZ) consists of two layers: (1) the parietal trophoblast giant cell (P-TGC), which is located the most peripherally, and (2) the spongiotrophoblast (SpT) layers. The P-TGC layer contains only p-TGCs, which exist directly on the surface of the maternal decidua and anchor the fetal-derived structures in the uterus [5]. These specific cells participate in the implantation of the embryo and in the processes of maternal adaptation to pregnancy, which occurs through an important endocrine activity. The SpT layer contains SpT and glycogen trophoblast (GlyT) cells, which play a major role in remodeling the decidual stroma and the maternal vascularization of the implantation site [2].

The placental labyrinth is a maze-like structure formed by interdigitated maternal blood sinusoids and fetal capillaries. In this region of the fetal–maternal interface, the fetal and maternal circulations are separated by a dynamic cellular structure, which is called the interhemal membrane (Figure 1C,E). It is composed of two continuous layers of syncytiotrophoblasts (SynT-Is and SynT-IIs) (fetal side) and a discontinuous layer of sinusoidal trophoblast giant cells (s-TGCs) (maternal side). Together with the endothelial cells of the fetal capillaries, they create a selectively permeable cellular barrier that is crucial for the exchanges of nutrients, gases, and waste product that occur between the maternal and fetal circulations [6]. This structural configuration of the fetal–maternal vascular interface of the mouse placenta is termed the *labyrinth type* (Figure 1D).

In humans, the definitive placenta becomes apparent around the 21st day of pregnancy. It is composed of two separate sheets, the chorionic plate and the basal plate, which surround the intervillous space as a cover and bottom, respectively (Figure 1A,B). The intervillous space of the human placenta is filled with maternal blood and contains the complex tree-like projections of the chorionic plate, which are called the chorionic villi [7] (Figure 1C). The chorionic villi are the structural and functional units of the human placenta. They contain capillaries that transmit blood from the fetal circulation. The specialized chorionic villi, by which the human placenta is attached to the uterine wall, are called anchoring villi. These anchoring villi attached to the decidua form proliferative cell columns giving rise to differentiated, extravillous trophoblasts (EVTs). Besides these anchoring villi, the floating villi are covered with multi-nucleate, terminally differentiated syncytiotrophoblast cells (SynTs) that are obtained from fusion of the sub-adjacent villous cytotrophoblast cells (CTBs, which support villi growth and regeneration) (Figure 1E). This double layer of cells which are bathed in maternal blood after establishment of the fetal–maternal circulation creates an interface for nutrient and gas exchanges between the maternal and fetal circulations. In contrast to the mouse placenta, this configuration of the fetal–maternal interface is classified as a *villous type* (Figure 1D).

Despite different morphological configurations, the functional interaction between the fetal and maternal vasculatures remains similar in both humans and mice. In both species, the maternal blood is in direct contact with the fetal-derived trophoblast cells. This type of placental organization is called “*hemochorial*”. However, on the basis of the number of layers of trophoblastic cells that are involved in the fetal–maternal interaction, the mouse placenta is defined as *hemotrichorial* (i.e., comprising three layers of trophoblast layers), whereas the human placenta is defined as *hemodichorial* (i.e., containing a double layer of trophoblast cells). Moreover, some elements of placental development between mice and humans remain similar. In particular, certain transcription factors involved in the expression of various placental genes have been described in both human and mouse placentas [8].

The labyrinthine arrangement of the capillary bed in the mouse placenta is also responsible for the completely opposite directions of fetal and maternal blood flow. The arrangement is classified as a *counter-current blood-flow exchanger* (Figure 1C). This is a highly efficient vascular system when compared to the less efficient *multivillous flow exchanger* of the human placenta, whereby maternal and fetal flows have the same direction. The efficiency of the murine placental vasculature consists in delivering an increased oxygen supply and increased passive diffusion potential. This allows a very rapid growth of the mouse embryo with a significantly small placenta compared to the human one. Indeed, the fetal-placental weight ratio is 20:1 in mice vs. 6:1 in humans. In addition, an effective placenta in mice also reduces the length of gestation: 19–21 days in mice vs. 37–42 weeks in humans.

It is believed that the villous-type placenta is less efficient but can compensate by increasing its mass. Moreover, the human villous placenta imposes lesser metabolic demands on the mother than the labyrinthine type. This enables a low daily fetal growth rate and a longer gestation for humans, which is advantageous given that they tend to produce larger offspring. Therefore, in humans, this exchange between villous trophoblasts and maternal blood represents an evolutionary compromise that helps to maintain the balance between maternal and fetal demands in longer gestations, without the depletion of maternal resources [9].

This phylogenic variability does not influence the reproductive qualities of the respective species due to the fact that they both fulfill their biological tasks and produce healthy offspring independently owing to the degree of simplicity or complexity of the placental structure.

Finally, the fetal vascular networks of both species contain nucleated red blood cells that result from fetal hematopoiesis, which occurs around the second month of human pregnancy and during E10.5-E11.5 of gestation in mice [3]. The presence of these nucleated red blood cells distinguishes the fetal circulation from the maternal blood flow, which contains exclusively non-nucleated red blood cells.

## 2. Comparative Anatomy of Fetal and Maternal Circulation in the Mouse and Human Placentas

In most mammals, including humans and mice, the spiral arteries (SAs) are the main sources of blood supply to the placenta and embryo (Figure 2). They originate from the uterine arteries (UAs), which branch into the arcuate arteries (AAs) and further into the radial arteries (RAs). The RAs cross the myometrium and, finally, end up in the spiral arteries (SAs), which form the very dense capillary plexus of the uterine endometrium [10]. During early mammalian pregnancy, the maternal spiral arteries, which are incredibly resistant, are transformed into large-caliber vessels that are capable of meeting the needs of the growing embryo. In both species, this process, which is called spiral artery remodeling (SAR), is mediated by specialized fetal-derived trophoblast cells that disrupt the smooth muscle layer and replace the pre-existing endothelial cells (see below). This results in a decrease in vascular resistance [11], an increase in vascular diameter, an increase in elasticity, and an induction of maternal blood strength and velocity. Although the SAR processes have the same rheological consequences, there are certain structural differences between mouse and human placentas.

In humans, the remodeled spiral arteries open into the intervillous spaces of the placenta, where the maternal blood passes over the surface of the placental villi, which also contain fetal capillaries. After maternal–fetal exchange and oxygen diffusion occur, the arterial pressure pushes the deoxygenated blood back into the uterine veins.

In contrast, in mice, several spiral arteries fuse into one to four centrally located channels that cross the junctional zone in order to generate the maternal part of the labyrinthine vascular network, so called the maternal sinusoids. In the labyrinthine region, the irregularly shaped maternal sinusoids interdigitate with the fetal capillaries (FCs) [1]. The oxygen-rich blood flowing through the maternal sinusoids is then collected and transmitted to the fetus through the umbilical cord vessels. The oxygen-poor blood is then collected by a few hundred vascular channels of the JZ, which converge to the large venous sinuses and, finally, form the uterine vein. Importantly, the distal parts of the spiral arteries with the maternal sinusoid and the vascular channels of the JZ are lined with fetus-derived trophoblastic cells, whereas the venous sinusoids are lined with maternal endothelial cells [2]. These specific subpopulations of trophoblast giant cells include: spiral-artery-associated trophoblast giant cells (SpA-TGCs), maternal blood-duct-associated trophoblast giant cells (c-TGCs), and s-TGCs, which line the spiral arteries, the maternal blood canal, and the maternal sinusoids, respectively. The expression of specific genes belonging to the prolactin and cathepsin gene families defines the subsets of these trophoblast giant cells [12] (see Table 1).

## 3. Brief Overview of Spiral Artery Remodeling (SAR)

During mammalian pregnancy, spiral artery remodeling (SAR) refers to the very complex physiological process that aims to establish structural and functional communications between uterine and placental vasculatures. During SAR, the uterine spiral arteries lose their muscular layers and become dilated. In addition, the inelastic tubes are also without maternal vasomotor control [13,14] (Figure 3). This allows the initial maximization of the placental perfusion and the maintenance of efficient diffusional exchange between the maternal and fetal circulations [15]. 

The SAR occurs at E8.5-10.5 [1] in mice and from 8 weeks onward in humans [10]. Importantly, the steps in the overall structural reorganization of the spiral arteries in the mouse placenta are quite comparable to those in humans, as they involve the interstitial and endovascular remodeling pathways [16,17,18]. Indeed, endovascular remodeling is coordinated by a specific subpopulation of fetal-derived trophoblast cells, which are called spiral-artery-associated trophoblast giant cells (SpA-TGCs) in mice [2,19] and extravillous trophoblasts (EVTs) in humans [20]. Both cells engage in invasive and migratory behavior. Moreover, they secrete high amounts of matrix metalloproteinases (MMPs) throughout the invasion by degrading the arterial basement membrane (i.e., elastolysis) [21] and induce the apoptosis of endothelial and smooth muscle cells and further displace them [22,23]. 

Interestingly, vasculogenic mimicry has also been suggested as a mechanism in humans [24]. It has, therefore, been proposed that endovascular trophoblasts may adopt the pseudo-endothelial cell phenotype during infiltration of the spiral arteries due to the fact that they share several phenotypic characteristics and regulatory factors with endothelial cells. As such, this should facilitate the replacement of maternal endothelial cells via fetal trophoblasts.

In addition to the dynamic modifications in uterine vascular organization, SAR also induces changes in uterine extracellular matrix composition [7] and immune cell populations [25]. This interstitial remodeling pathway is also mediated by glycogen trophoblast cells (GlyTs) and extravillous trophoblasts (EVTs) in mice and humans, respectively. They secrete large amounts of MMPs and simultaneously remodel the parenchymal compartment of the decidua throughout the penetration of the uterine wall. After this, they reside in the perivascular regions of the decidual vessels and participate in their dilation [26] (Figure 4).

Recently, transcriptomic profiling of the human placenta has allowed the characterization of additional types of invasive trophoblasts which infiltrate the uterine veins, lymphatic vessels, and endometrial glands [27]. These additional types of invasions participate in the parenchymal reorganization of the uterus, as well as in the histiotrophic nutrition of the embryo. In addition, they contribute to the drainage of deoxygenized blood and waste products from the intervillous space toward the maternal circulation [28]. Nevertheless, the precise mechanisms of their regulation, as well as of their mouse analogues, are still unknown.

Altogether, spiral artery remodeling is a complex physiological process that involves time-coordinated crosstalk between uterine vessels and fetal-derived trophoblast cells, which ensures the establishment of an appropriate exchange apparatus for the transfer of oxygen, nutrients, and waste products between the maternal and fetal circulations. Despite similarities in the general mechanisms of the RAS, several peculiarities between humans and mice have also been reported. The main comparative characteristics, as well as the phenotypic profiles of the participating cells, are summarized in Table 1.

**Table 1 cells-12-00620-t001:** Comparative characteristics of human and mouse spiral artery remodeling.

HUMAN		MOUSE
**Common features**	Spiral artery remodeling (SAR) is mediated by trophoblast cell populations [1,10,29]; This includes the endovascular and interstitial pathways of vascular remodeling [7,27]; Trophoblasts invade the spiral arteries and localize to the vessel lumens, thus forming the cellular plugs (human) or intraluminal infiltrations (mouse) [10,16]; The remodeling induces fluid shear stress on the vascular endothelium [30]; Trophoblasts secrete different MMPs throughout the remodeling [21,31,32]; Trophoblast cells progressively replace the underlying endothelial and muscular cell layers [16,18,31,32]; They acquire the pseudo-endothelial phenotype by expressing endothelial cell adhesion molecules and by secreting angiogenic factors [24,33]; The molecular pathways of SAR are conserved between mice and humans (involve similar transcription factors) [8]; SAR is also supported by uterine natural killers (uNKs) [34,35,36]; Remodeling results in the loss of the spiral artery muscular layer, elasticity, and maternal vasomotor control [15,37]; Remodeled SAs are highly dilated arteries with low resistance [11,15,18]; The interstitial trophoblasts spread throughout the uterine stroma after invasion [27].	
**Distinctive features**	■ SAR is mediated by extravillous trophoblasts [7,10]; ■ The placental implantation occurs up to the inner third of the myometrium [38]; ■ The interstitial trophoblasts spread throughout the uterine stroma after invasion (i.e., in the inner third of the myometrium) [27,39]; ■ The remodeling of the decidual and myometrial segments of SAs [10,40]; ■ Additional endovenous, endolymphatic, and endoglandular trophoblastic invasions occur [39].	■ SAR is mediated by SpA-TGCs (endovascular) and GlyTs (interstitial) [41]; ■ The placental implantation is restricted to the decidua [42]; ■ The invasion of interstitial and endovascular trophoblasts is superficial [1,2]; ■ The remodeling exclusively concerns the decidual segments of SAs [43]; ■ Not documented in mice.
**Cell-specific markers and secretory phenotype**	**EVT**-HLA-G [44]; CK7, CK8, and CK18 [39]; HSD3β1; β-Hcg; MMP-1, MMP-2, MMP-9, MMP-7, MMP-3, and TIMP-1/TIMP-2 [25,45]; **uNK**-CD56 ^bright^, CD16 ^low^, and CD3(-) [36]; Dolichos biflorus (DBA) lectin; IFNγ, TNFα, and TGF-β1; MMP-2, MMP-7, MMP-9, and TIMP-1/-2/-3 [34,35,36].	**SpA-TGC**-*Prl2c* (*Plf*), *Prl4a1* (*PlpA*), *Prl7b1* (*PlpN*), *Cts8*, *Rgs5*, and *Lnc7* [3,5]; **GlyT***-Pcdh12*, *Cx31*, and *Prl7b1; PAS* *MMP 9 and Decorin (Dcn)* [5,26]; **p-TGC**-*Prl3d1(Pl-I), Prl3b1(Pl-II) (all),* and *Prl4a1 (PlpA) (secondary)* [5]; **uNK**-*DBA lectin and Periodic-Acid Shiff (PAS),* CD3(-)/CD122(+), and *IFNγ* [46].

## 4. Overview of Estrogen Actions during Pregnancy

Sex steroid hormones and, in particular, estrogens are essential regulators of reproductive competence. They perform pleiotropic biological actions during pregnancy. The most important of these are summarized in Table 2.

Estrogens are steroid hormones, which are secreted by all vertebrates, and are indicative of a common origin and involvement in important endocrine functions (reviewed in [47]). In non-pregnant women, estrogen fluctuates during the menstrual cycle and signals the anterior pituitary in both a feedback and feed-forward loop to stimulate LH and FSH and cause ovulation. Indeed, four major, naturally occurring estrogens are characterized in women: estrone (E1), estradiol (E2), estriol (E3), and estetrol (E4) [48,49]. Their levels of secretion in different species vary according to age (reproductive/menopause) and physiological condition (non-pregnancy/pregnancy/lactation). In non-pregnant females, estrogens are principally secreted by the ovaries. However, smaller amounts are also produced by the adrenal glands and adipose tissues [50].

Endocrine function differs in human and mouse placentas [50]. In mice, the corpus luteum is required to produce progesterone throughout gestation. The second difference is that the genes encoding enzymes that are involved in steroidogenesis are not expressed in the mouse placenta during the second half of gestation, whereas they are expressed in humans, albeit late in the gestation period. Accordingly, although ovaries are generally the main producers of hormones, the human placenta becomes the main organ producing estrogens during pregnancy [51,52]. This occurs after the 9th week of human pregnancy [53], through the suppression of the secretion of ovarian estrogens. This results in a progressive increase in their circulating levels after mid-gestation [49,54,55]. Importantly, placental estrogen biosynthesis is unique to humans and higher primates [56], and is mediated by the aromatization of fetal dehydroepiandrosterone sulfate (DHEA-S), which is secreted by the fetal adrenal gland [57,58]. After hydroxylation in the fetal liver, adrenal DHEA-S reaches the placenta, where it is cleaved and converted into androstenedione by the abundantly expressed 3βHSD-I enzyme. As the human placenta also expresses a higher quantity of CYP19 enzymes (aromatases), androstenedione is then further converted into E1, E2, and E3 [59,60]. Notably, E3 is mainly secreted by the placenta during pregnancy [60,61]. Due to the hemochorial nature of the placenta, more than 90% of the E3 that is formed in syncytiotrophoblasts enters the maternal circulation [61]. E4 is another natural estrogen produced exclusively during pregnancy by the fetal liver, albeit only in humans and higher primates. E4 is present as early as at 9 weeks of gestation, with relatively high levels in the fetus and lower levels in the maternal circulation [62,63]. While the functional and physiological roles of E3 and E4 are not fully understood, we will discuss here the roles of E2 during pregnancy, which regulates a myriad of functions during human and non-human pregnancies [49]. The maternal plasma levels of E2 increase progressively during pregnancy from 20 to 500 pg/mL (in non-pregnant women, depending on the menstrual cycle), up to 10,000–40,000 pg/mL at term [55]. In mice, E2 levels gradually increase from 3 to 15 pg/mL (diestrus–proestrus phase), up to more than 100 pl/mL at the end of the pregnancy [64,65]

In addition, aromatase activity is also present in uterine stromal cells [66] and arterial smooth muscle cells [67,68] during pregnancy, thereby suggesting the additional local secretion of estrogens. E2 plays an important role in reproductive organogenesis and the development of secondary sexual characteristics. It regulates the cyclic patterns of uterine cell proliferation and vascular development that occur throughout the menstrual cycle. The actions of estrogens are also critically important for normal reproductive behavior, as well as the establishment and maintenance of pregnancy [69].

At the molecular level, the predominant biological effects of estrogens are exerted by two main receptor subtypes that include ERα and ERβ, which mediate estrogenic effects in target tissues. Following the binding of estrogens, these receptors act as transcription factors and regulate numerous genes in a tissue- and cell-specific manner. These actions are described as the nuclear actions of ERs. In addition to these nuclear actions of receptors, a fraction of ERα is expressed at the plasma membrane due to its palmitoylation on cysteine 451. This membrane ERα expression triggers rapid signaling via transient activation of several kinases, which then phosphorylate nuclear ERα in order to induce the expression of many genes. These recently described “non-nuclear” effects have been attributed to the cell-membrane-initiated steroid signaling (MISS) of ERα [70]. These receptors are then present in multiple tissues and species. Both ERα and ERβ are present in uterine vascular endothelial cells during pregnancy [71]. Additionally, ERα expression is also found in mesenteric arterial smooth muscle cells, where it mediates pregnancy-associated vasodilatation [72], thereby suggesting that both cell types are targets for elevated estrogen during pregnancy.

Although E2 is a fundamentally important driver in preparing the mammary gland for lactation and parturition, the primary role of this steroid hormone in mammalian pregnancy is the stimulation of uterine vascular growth and increase in uterine blood flow [73]. Estrogen directs utero-placental circulation by exploiting different mechanisms: initially by stimulating uterine angiogenesis [74,75], which then promotes vascular remodeling, and, finally, by controlling hemodynamics [76].

### 4.1. Estrogen Actions during Embryo Implantation and Uterine Angiogenesis

It is now well-documented that the steroid hormones 17β-estradiol and progesterone function via their respective receptors, which orchestrate the differentiation and remodeling of uterine tissues in order to prepare them for embryo implantation and pregnancy establishment. Several studies have demonstrated that estrogen optimizes the intrauterine environment for decidualization and embryo implantation in early pregnancy [77,78,79]. This process is mediated by the uterine-compartment-specific actions of ERα. In addition, endometrial stromal ERα apparently plays a crucial role in the estrogen-induced proliferation of uterine epithelial cells, while epithelial ERα, in turn, plays a pro-survival role in embryo implantation [80]. Indeed, uterine epithelial ERα has also been shown to control decidualization through the direct epithelial–stroma dialogue that occurs during pregnancy establishment [81]. In agreement with these data, a small estrogen surge occurring during the “implantation window” is essential for the uterine receptivity to the embryo. Interestingly, while a transient rise in estrogen precedes embryo implantation in mice (E4.5), these changes also coincide in women (~7–10 d after ovulation) [77]. Once implantation is complete, angiogenesis and further remodeling of the uterine vascular bed take place.

**Table 2 cells-12-00620-t002:** **Main biological functions mediated by estrogens during pregnancy**. H: human; M: mouse; P: primate; R: rat; EC: endothelial cell; SMC: smooth muscle cell; OXTR: oxytocin receptor; CX43: Connexin43; PG(R): prostaglandin (receptor).

Target		Function	Species	Implicated ER	Ref.
** Implantation **	**Uterine epithelium**	Tissue responsiveness and functioning;	H, M	ERα	[80]
	**Uterine stroma**	Epithelial growth;	H, M	ERα	[82,83]
		Pre-implantation remodeling;	H, M	ERα	[80,83,84]
	**Blastocyst**	Attachment/implantation;	H, M	ERα	[77,78,79]
	**Decidua**	Decidual angiogenesis.	H, M, P	ERα	[51,74,85]
** Vascular remodeling **	**Trophoblast**	Viability;	H		[83,86]
		Proliferation/differentiation;	H	ERα, ERβ	[87,88]
		Invasion;	H, P	ERα, ERβ	[83,86]
	**ECM**	MMP-dependent remodeling;	H		[89,90,91]
	**Vascular endothelium**	NO and VEGFA synthesis;	H, M	ERα	[92,93]
	**uNK**	Migration and functional activity.	H		[36,94,95]
** Systemic hemodynamics **	**Uterine blood flow** **Umbilical artery** **Myometrial artery** **Placental artery** **Mesenteric artery** **Aorta** **Heart**	Increase arterial wall diameter;	H, M	EC/SMC ERα, ERβ	[51,96,97],
		Relaxation;	H	EC ERα, ERβ	[98]
		Relaxation;	H	EC ERβ	[99]
		Relaxation;	H	SMC ERα	[99]
		Relaxation;	R	EC and SMC ERα	[100]
		Relaxation;	R	EC and SMC ERα	[72,100]
		No major influence;	H, M		[97,101]
		Cardio-vascular adaptation.			
** Labor **	**Cervix**	Softening and dilation;	H, M		[102,103,104]
	**Myometrium**	Contraction increase;	H, M		
	**Ovary**	Luteolysis.	H, M		

In many mammalians, the implantation of blastocysts into the uterus induces a massive increase in the decidual microvasculature in order to meet the requirements of the growing embryo. This process, which includes the growth of uterine arterioles from the pre-existing uterine vasculature, is called decidual angiogenesis. This decidual angiogenesis is governed by several local and systemic factors, such as vascular endothelial growth factor (VEGF), placental growth factor (PGF), angiopoietins (Angs), and their respective receptors [105,106,107]. All these factors are essential determinants of decidual angiogenesis. Furthermore, as mice are deficient in these pro-angiogenic factors, they exhibit defective placental and embryonic angiogenesis, which lead to severe developmental abnormalities and embryonic lethality [108]. The main sources of these factors are as follows: uterine endothelial, stromal, epithelial, and uNK cells [35,36]. It is important to note that all of the above cells are sensitive targets of estrogen (see below).

Numerous studies have shown that, in humans as in mice, E2 is the major angiogenic hormone throughout the menstrual cycle and is present at the onset of placentation. In both species, the crucial role in this process has been attributed to the locally secreted placental E2 [51,74,85], which induces the expansion of the post-implantation vascular bed by stimulating the secretion of VEGF, Ang2, HIF-2α, and many other pro-angiogenic factors [109].

The major mediator of the effect of estrogen during uterine angiogenesis remains VEGF. VEGF, which is secreted in response to E2, activates its receptor, VEGFR-2, on endometrial endothelial progenitors and mature endothelial cells. As such, it thus promotes their proliferation and migration [110]. The higher growth response and increased responsiveness to VEGF is most likely due to an increased affinity or number of VEGF receptors on endothelial cells. However, E2-induced increase in VEGF synthesis is also found in villous and extravillous cytotrophoblasts [88], as well as in uterine epithelial cells [111], stromal fibroblasts [112], and mastocytes [99]. All these cells are, then, additional targets for estrogen-induced angiogenesis. Indeed, the modulation of vascular function is also mediated by other pro-angiogenic factors, including CCL2, which are secreted by human uNKs in response to estrogen [94].

In contrast, the precise mechanism of PGF-regulated angiogenesis is not well understood. It is believed that PGF amplifies the responsiveness of ECs to VEGF during vascular network formation, which is due to the fact that they share the activating receptor VEGFR-1 [106]. Importantly, the proangiogenic stimuli of PGF and VEGF bind to soluble (s)FLT1 in order to prevent excessive angiogenesis [113]. Furthermore, a correlation between (s)FLT1 secretion and E2 concentration has been demonstrated [114]. Therefore, this may be an additional mechanism by which estrogen may control the degree of expansion of the uterine vascular bed.

Effective angiogenesis is further mediated by hypoxia-inducible family factor 1 (HIF-1), the level of which increases during the hypoxic period in early gestation. It stimulates the transcription of several angiogenic genes and induces the production of several cytokines and growth factors, including VEGF [115]. HIF-1 also influences uterine vascular permeability, regulates the cell cycle, and defines trophoblast cell fate. It must be noted that ERα signaling has been shown to regulate the HIF-1 pathway under different physiological and pathological conditions [116,117].

In addition to directly promoting uterine angiogenesis via endothelial cell proliferation and migration, E2 also increases vessel permeability and induces vasodilation [93]. The most important molecule supporting this process has been attributed to nitric oxide (NO), which is synthesized via uterine endothelial cells, endometrial stromal cells, and myometrial smooth muscle cells, in response to E2 [118]. Locally secreted NO induces vasodilation and thus causes a perfusion of the uterine arterial bed [119]. In addition, it synergizes the pro-angiogenic actions of VEGF [120]. E2 also indirectly supports decidual angiogenesis. It stimulates smooth muscle cells [121] and immune cells in order to increase the local synthesis of several growth factors, cytokines, and vasoactive mediators [122].

During the first trimester of gestation in the baboon, the placental expression of many other growth factors increases dramatically in order to facilitate the coordinated development of the vascular system through the sprouting and elongation of the placental villi. Indeed, Robb, Albrecht, and Pepe [123] demonstrated that placental vascularization was not altered by P450 aromatase inhibition, despite the decline in estrogen and, consequently, the decrease in vascular endothelial growth/permeability factor levels. The underlying reason(s) for this is unknown, but it has been strongly suggested that the compensatory synthesis of other angiogenic factors, e.g., PGF, may support angiogenesis. Another hypothesis is that other cells in the villous placenta may become an important source of VEGF/PF with estrogen deprivation. This likely occurs in order to maintain angiogenesis, which is possibly secondary to the induction of hypoxia and possesses a well-established role in terms of stimulating VEGF expression in various tissues.

In summary, proper uterine angiogenesis, which is largely influenced by E2, requires the balanced secretion of pro- and anti-angiogenic factors in order to achieve a stable and mature decidual vascular network, which is essential for nutritional supply during feto-placental development.

### 4.2. E2 and Hemodynamics

In most mammalian pregnancies, the progressive increase in E2 levels coincides with the several-fold increase in utero-placental blood flow. This increase in utero-placental blood flow is required for the delivery of adequate oxygen and nutrients that are necessary for fetal and placental growth and development and for normal pregnancy outcomes. In pregnant women, insufficient blood flow is associated with pregnancy complications, such as pre-term delivery, intrauterine growth retardation, or even pre-eclampsia, which is only observed in humans. Hemodynamic changes are principally affected by a profound decrease in vascular resistance, which is principally due to a combination of expansive remodeling (vessel growth) and increase in vasodilation (reviewed in [76,96,124]).

#### 4.2.1. Role of Estrogens in Uterine Arterial Remodeling 

This vascular process corresponds to the adaptation of the arterial diameter and wall composition to meet the increased blood-supply requirements under different physiological conditions. In pregnant rats, a greater than four-fold increase in uterine artery weight during the middle of pregnancy was observed with an even greater increase toward the end of pregnancy [100]. The diameter of the human uterine artery is approximately doubled [125], and similar changes have been reported in rodents. An important role for female sex steroid hormones (progesterone and estrogens) in this uteroplacental vascular adaptation during pregnancy has been documented in a number of studies (reviewed in [96]). Indeed, hormone replacement in ovariectomized mice and surgical ligation, induction of pseudo-pregnancy (conditions that can be induced by sexual stimulation that increases sex steroid levels similar to the levels seen in early pregnancy) demonstrate that steroids may initiate the process of uterine vascular remodeling, inducing an increase in lumen diameter, cross-sectional area, and proliferation of uterine medial SMCs [14,126,127]. E2 regulates local uterine blood flow by combining its actions on endothelial and smooth muscle cells, as both express ERs [72,76,128].

#### 4.2.2. Role of Estrogens in Uterine Vasodilation 

E2 also largely contributes to pregnancy-associated uterine vasodilation by acting both on uterine arteries and on myometrial and placental arteries [98,99,100] via rapid synthesis of NO [129,130]. NO is a powerful vasodilator, and pregnant rats treated with L-NAME, as well as eNOS-knock out mice, showed reductions in vascular outward remodeling [92]. Interestingly, the relaxation of the myometrial uterine arterial segment is endothelial NO-dependent and involves the activation of both ERs, whereas the placental artery is relaxed in an endothelial-independent manner and requires the selective activation of ERβ [99]. Notably, the acute production of the local vasodilator NO by endothelial cells following E2 treatment involves the non-genomic effects of E2, which are mediated, in part, by an ERK pathway and PI3K/PKB activation, possibly via plasma-membrane-localized estrogen receptors [130].

However, a more recent study assessing ex vivo uterine arterial dilation in response to applied pressure did not demonstrate any alteration to pregnancy-associated uterine arterial remodeling in C45A-ERα mice deficient in membrane ERα signaling at E9.5, while C451A-ERα mice clearly harbor a defect in rapid NO synthesis [131,132]. Thus, this study strongly indicates that membrane-initiated ERα signaling is not required for arteriolar remodeling in uterine arteries in response to the physiological increase in blood flow during pregnancy. This is most likely due to the fact that the contribution of NO to these changes is modest, because the uterine arterial remodeling is altered but not absent in pregnant Nos3 knock-out mice [92].

#### 4.2.3. Role of E2 in Relation to Myogenic Tone

The E2-induced changes in uterine blood flow also underlie the adaptation of the arterial muscular layer. E2 can directly regulate myogenic activity and the contractility of uterine vascular SMCs, without the involvement of ECs and NO. Selective modifications of the intrinsic properties of SMCs induce changes in the myogenic tone of the uterine arteries, thus leading to a reduction in vascular resistance and arterial pressure, as well as ultimately contributing to an increase in uterine blood flow [133,134].

In addition to local vasodilator effects, E2 also influences systemic cardio-vascular adaptation and also results in changes in fluid volume that normally occur [76,97]. Indeed, the prolonged systemic chronic infusion of E2 reproduces cardiovascular adaptations that are associated with pregnancy, including increases in heart rate and cardiac output, systemic vasodilation, and attenuated pressor responses to infused angiotensin II [75].

In summary, the proper development of the utero-placental vasculature ensures an uninterrupted blood supply to the growing embryo. However, after the establishment of blood flow, the maternal vasomotor control becomes the predominant regulator of the utero-placental blood circuit. This is achieved by the combined actions of the maternal cardiovascular system, hormones, and local vasoactive mediators [129].

### 4.3. Estrogen Actions during Spiral Artery Remodeling

Experiments with different animal models have suggested that estrogens can control the viability, differentiation, proliferation, and invasion of trophoblast cells during utero-placental vascular morphogenesis [86,87,88,135].

#### 4.3.1. Role of Estrogens in SAR in Primates 

Direct regulation of trophoblast biology by fluctuating serum E2 concentration has been relatively well-characterized in primates [29,88]. A lower E2 level has been shown to promote trophoblast invasion, whereas a higher concentration inhibits this process [74]. Moreover, a high serum E2 level was shown to induce morphological differentiation of placental villous trophoblasts [136] and migration of EVTs [88]. Interestingly, a more recent study from Aberdeen et al. demonstrated that premature elevation of E2 levels suppresses trophoblast cell invasion into the spiral arteries, resulting in defective maternal vascular remodeling [137]. Therefore, it has been suggested that the progressive increase in circulating E2 levels throughout pregnancy controls the extent of spiral artery remodeling in non-human primates.

#### 4.3.2. Role of Estrogens in SAR in Humans

Several in vitro and ex vivo studies have documented that E2 influences the behaviors of human syncytiotrophoblast, cytotrophoblast, and EVT cells via ERα and ERβ, although the precise mechanisms are not fully understood [87,135,138]. Estrogen was shown to stimulate the differentiation and invasion of cytotrophoblasts [83,86], as well as the migration of extravillous trophoblasts in a dose-dependent manner [89,90]. In this process, the primary role has been attributed to a locally synthesized placental E2 that modulates the secretory activity of invading trophoblasts to promote interstitial remodeling and migration. Interestingly, the premature elevation of E2 circulating concentration above the upper range alters trophoblast survival through its pro-apoptotic and anti-proliferative influence, thereby suggesting a possible interplay between estrogen concentration and placental morphogenesis [139].

Other studies have documented that E2 additionally promotes the proliferation and migration of human villous fibroblasts and vascular pericytes via ERα during SAR [140]. It also stimulates the production of uterine vascular MMPs [91], NO [92], and VEGF [93] in different cells of the utero-placental unit. In addition, E2 secreted by maternal decidual cells was demonstrated to regulate the recruitment, migration, and secretory activity of uNKs—well-recognized key regulators of SAR [34,36,94,95,141]. This is not surprising, as uNKs are immunopositive for ERβ but negative for ERα, whereas both receptors are expressed by maternal decidual cells [142]. This divergent pattern of estrogen receptor expression must underlie the interaction between these different cellular compartments during early placentation.

#### 4.3.3. Role of Estrogens in SAR in Mice 

Although there is compelling evidence that uNKs also play important roles in murine uterine vascular remodeling [143], this process appears to be independent of E2, since mouse uNKs lack ERα and ERβ expression [144]. Thus, it is thought that E2 is not absolutely necessary for the differentiation and activity of uNK precursor cells during vascular remodeling in mice.

It should also be noted that, in contrast to humans, the expression of estrogen receptors by trophoblasts has never been reported in mice. Moreover, the role of estrogen receptors in murine spiral artery remodeling has received limited attention, given that all mouse models invalidated for ER isoforms (both ERα and ERβ) or its sub-functions (ERα-AF1° and ERα-AF2°) are infertile [70]. Thus, the C451A-ERα mouse [131], mutated at the palmitoylation site of ERα to specifically abrogate ERα membrane expression, has appeared as the first mouse model with which to study the effects of maternal ERα on trophoblast activity during early placentogenesis in vivo. Indeed, C451A-ERα mice show several gestational abnormalities, including parturition failure, leading to total neonatal lethality of the offspring [132]. We demonstrated that the partial embryonic death appearing at about E9.5 in mutant females resulted principally from the impaired expansion of *Tpbpa*-positive spiral-artery-associated trophoblast giant cells (SpA-TGCs) to the utero-placental unit. This observation was associated with a significant decrease in the thickness of the utero-placental unit and a reduction of the total placental surface, confirming inadequate remodeling of spiral arteries. In addition, and most importantly, we showed that *Tpbpa*-expressing SpA-TGCs were completely absent from the placentas of dying embryos of mutant mice, which absence was accompanied by significant structural and vascular disorganization.

These data strongly indicate that the maternal Estrogen Receptor ERα is important in controlling the expansion of specific trophoblast cells during spiral artery remodeling in mice. However, the reduced trophoblastic activity reported in this work obviously raises the question of the identity of the maternal cells responsible for this phenomenon, primarily because C451A-ERα mothers exhibited increased circulating levels of E2 and progesterone throughout pregnancy. Given the direct regulatory role of E2 in trophoblast invasion in primates, it is difficult to conclude what the predominant trigger for the reduced expansion of SpA-TGCs in the conception of C451A-ERα mice is. Is it the result of elevated maternal E2 that suppresses trophoblast invasion through the spiral arteries, or is it a consequence of loss of membrane ERα function in the maternal environment that impacts fetal and maternal cell communication and thus impairs the differentiation of SpA-TGCs from progenitor cells, or it is a combination of all of these factors? These interesting observations deserve further investigation to fully characterize the estrogen-induced mechanisms of altered trophoblast biology during early placentogenesis in mice.

## 5. Estrogens and Abnormal Placentation

Several studies have highlighted the role of estrogens in the pathophysiology of several pregnancy-associated complications, such as pre-term delivery, intrauterine growth restriction, and pre-eclampsia [66,145,146,147,148]. Maternal blood concentration of E2 was found to be significantly positively associated with birth weight and perhaps placental weight, while total gestational duration was inversely associated with E2 [149]. Furthermore, while normal E2 levels are important to maintain uterine receptivity for blastocyst implantation [79], a high serum E2 level has been reported to impair implantation in humans [150], as well as in mice [151].

Interestingly, women with polycystic ovary syndrome who have dysregulated estrogen and androgen secretion exhibit placental vascular abnormalities, including reduced endovascular trophoblast invasion and disturbed uterine–placental blood flow [152]. Importantly, the placentas of these patients also express higher levels of ERα [153]. Based on these observations, increased expressions of placental estrogen and androgen receptors were also found in prenatally androgenized rats [154]. Altogether, these data suggest an increased sensitivity of the placenta to estrogens that could directly or indirectly modulate placental vascular morphogenesis.

Furthermore, in human pathological pregnancies, which are associated with placental aromatase deficiency, both mothers and fetuses demonstrate severe virilization due to androgen excess. It has, therefore, been proposed that placental estrogen is synthesized in order to regulate the metabolism of androgens into estrogens and thus protect the fetus and mother from virilization [73,155].

Indeed, a central role in the pathogenesis of these diseases has been attributed to the “shallow” invasion of trophoblast cells, which results in the restricted formation of arterioles that are highly resistant and hypertrophic [156]. Moreover, clinical studies in patients with pre-eclampsia have correlated poor development of placental vasculature with lower circulating levels of E2 [52,87,157]. Indeed, lower estrogen levels can be due to the abnormal activity of the enzymes that are involved in their biosynthesis [147]. Moreover, the ultrastructural study of human placentas by Bukovsky et al. demonstrated that placentas with an abnormal morphology lack ERα expression in contrast to normal placentas, thereby suggesting the considerable impact of estrogens on placental morphogenesis [140].

Other etiological mechanisms of abnormal utero-placental vascular remodeling have also been described in patients with pregnancy complications. These include a reduction in the number and invasive capacities of trophoblasts [17,158], impairment of their metalloproteinase activities [159,160], altered uNK numbers and activation [44], and inadequate expression of adhesion molecules in trophoblast cells [161], thereby leading to a failure to acquire the endothelial phenotype [162]. These abnormal findings cannot be directly attributed to dysregulated estrogen signaling, as their circulating levels and ER expression were not demonstrated in these studies. However, as all of these processes are strongly regulated by estrogens during normal placentogenesis, we can assume that they may exert indirect modulatory effects in response to estrogen during pre-eclampsia.

The crucial role of trophoblasts in the pathogenesis of pregnancy-associated complications has also been demonstrated in mice. A mouse model lacking *Tpbpa*-positive progenitor cells, which give rise to SpA-TGCs, demonstrated narrower spiral artery formation due to insufficient vascular remodeling associated with increased embryonic lethality [20]. Unfortunately, steroid hormone levels in these mice were not reported. In addition, mice deficient in catechol-O-methyltransferase, which converts 17β-triol to 2-methoxyoestradiol, show symptoms of pre-eclampsia, thus supporting a causal role for reduced estrogen in pregnancy complications [163]. Furthermore, in the rat model of pre-eclampsia, exogenous E2 administration reverses the major alterations, including blood pressure and proteinuria. It also improves neonatal mortality and fetal weight, and it decreases inflammatory response and endothelial damage [164].

## 6. Conclusions

In conclusion, estrogen signaling in utero-placental vascular morphogenesis remains a fascinating but understudied area of research. Moreover, it is a critical process, alteration to which may compromise the wellbeing of the mother and the development of the fetus. Although mouse and human placental structures are different, mouse models have been instrumental in studying the physiology and pathophysiology of mammalian pregnancy. This is due to their reproductive advantages, the relative simplicity of genetic manipulation, the similarities between mouse and human placental development (see Table 1), and ethical and cost considerations. Studies in knockout mice have also significantly improved our understanding of the cellular and molecular mechanisms of placentation and the consequences of their dysregulation. However, whilst keeping in mind the differences between mice and humans, care must be taken in extrapolating physiopathological mechanisms underlying utero-placental vascular morphogenesis and hemodynamics from mice to humans: (i) primarily, due to the certain unique characteristics of the mouse placenta (as summarized in Table 1), but also (ii) because the complications associated with pregnancy, such as pre-eclampsia and intrauterine growth retardation, are human-specific conditions that do not naturally occur in mice.

Nevertheless, although certain pathological mechanisms may not be directly applicable to humans, the various mouse models of pregnancy may shed new light on the fundamental mechanisms of maternal adaptation that are regulated by estrogen during gestation. This could lead to the identification of therapeutic targets or even preventive strategies for pregnancy disorders and, consequently, could reduce perinatal mortality and morbidity.

## Figures and Tables

**Figure 1 cells-12-00620-f001:**
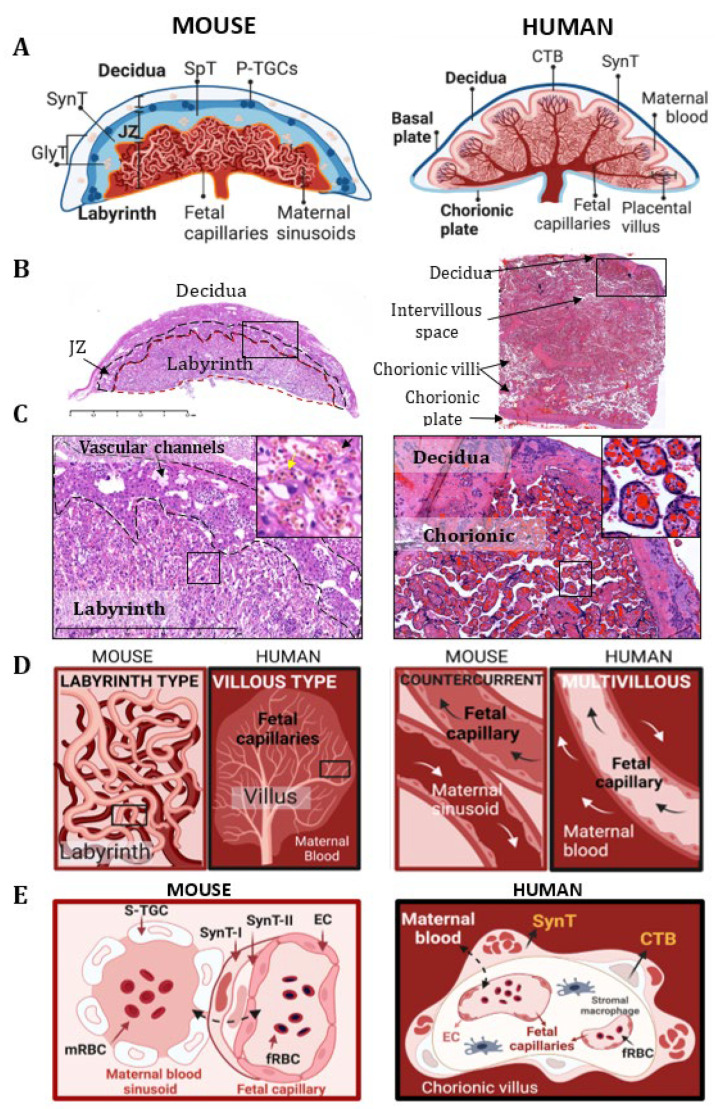
Structural and functional characteristics of mouse and human placentas (**A**,**B**). Schematic representation (**A**), with H&E-stained cross sections (**B**) of mature mouse (E14.5) and human placentas (40 weeks). Mouse placenta contains three main layers: the maternal decidua (Dec), the fetal-derived junctional zone (JZ), and the labyrinth (Lab). The JZ and labyrinth are respectively delineated by black and red dotted lines in the histological sections. The JZ is composed of GlyTs (glycogen trophoblast cells), SpTs (spongiotrophoblast cells), and P-TGCs (parietal trophoblast giant cells). The labyrinth consists of an anastomosing network of maternal blood sinusoids and fetal capillaries, which are surrounded by SynT cells (syncytiotrophoblasts). Scale bars: 2,5 mm (left) and 1 mm (right). The human placenta is composed of two different surfaces, the fetal surface (or chorionic plate) and the maternal surface (or basal plate), which are attached to the maternal decidua. Indeed, multiple chorionic villi, when arising from the chorionic plate, contain fetal capillaries and are surrounded by SynT and CTB (cytotrophoblast) cell layers. Moreover, maternal blood circulates in the intervillous space and on the surface of each chorionic villus. (**C**) Higher magnifications of the histological sections of the mouse placenta (left), the vascular channels of the JZ (arrow), and the labyrinthine vascular network, consisting of maternal and fetal interdigitated vasculature. The magnified view of the fetal–maternal vascular interface (on the right corner) shows nucleated fetal-derived red blood cells in the fetal capillaries (yellow arrow) and non-nucleated red blood cells in the maternal circulation (black arrow). Scale bars: 1 mm and 50 µm. In the human placenta (right), the fetal capillaries are included in the chorionic villi. Each villus is surrounded by maternal blood, which circulates in the intervillous spaces. In the human placenta, when at term, nucleated red blood cells are present until the end of the first trimester. Thereafter, their presence decreases, such that they are not apparent at term. Scale bars: 200 µm and 50 µm. (D) The comparative structural and functional features of mouse and human placentas, thereby demonstrating the labyrinthine (mouse) and villous (human) organizations of fetal–maternal interfaces. The mouse labyrinth represents a network of direct anastomoses of maternal and fetal vasculature, whereby the blood circulates in the countercurrent direction. In regard to the human placenta, the fetal–maternal interface is represented by fetal capillaries that are located in the chorionic villi and the surrounding maternal blood. The flow between the fetal and maternal circulation in the human placenta is classified as a multivillous flow organization. (E) The comparative organization of the cellular barrier between the maternal and fetal blood compartments is called the interhemal membrane. In the mouse placenta, the maternal blood, which flows through maternal sinusoids, is separated from the fetal blood, which flows in the fetal capillaries, by: a perforated layer of sinusoid trophoblast giant cells (S-TGCs), two continuous layers of syncytiotrophoblast cells (SynT-Is and SynT-IIs), and fetal endothelial cells (hemotrichorial placenta). The fetal capillaries are lined with endothelial cells, while the maternal sinusoids are surrounded by S-TGCs of fetal origin. In the human placenta, a layer of multinucleated syncytiotrophoblast (SynT) cells and an underlying layer of mononucleated cytotrophoblast (CTB) cells cover the villi and prevent direct contact between the maternal blood and fetal blood capillaries (hemodichorial placenta). The arrows indicate the directions of exchange between the two circulations. fRBCs: fetus-derived red blood cells; mRBCs: maternal red blood cells; ECs: endothelial cells.

**Figure 2 cells-12-00620-f002:**
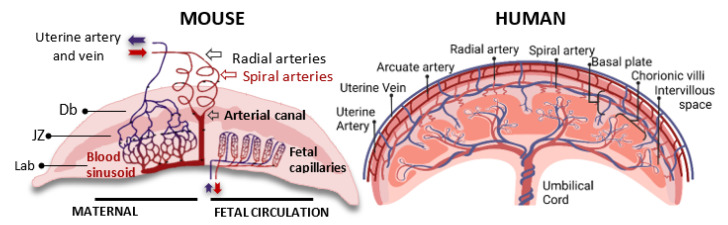
Organization of the utero-placental vascular system in mice and humans. In mice, as in humans, the placental vascular tree originates from the highly resistant uterine arteries (UAs), which branch into radial arteries (RAs) and then terminate in the spiral arteries (SAs). The SAs supply the decidua at the implantation site. Moreover, in the mouse placenta, the SAs fuse in order to form the centrally located arterial canals (CCs), which supply maternal blood to the fetal compartments of the placenta. The CCs enter the junctional zone (JZ) and generate maternal arterial sinusoids in the labyrinth. Moreover, the maternal sinusoids interdigitate with the dense capillary network of the fetal vasculature and thus create the interface for nutrient and gas exchange. The oxygen-poor blood flow returns to the vascular channels of the JZ, which become venous sinusoids just above the JZ and terminate in the uterine vein. In this simplified diagram, part of the maternal circulation is shown on the left. In addition, part of the fetal circulation is illustrated on the right. In humans, the spiral arteries open into the intervillous spaces of the placenta where the maternal blood passes over the surfaces of the placental villi, which contain fetal capillaries. After maternal–fetal exchange, blood from the intervillous space is drained by the utero-placental veins, the openings of which are on the floor of the intervillous space. The diagram on the left was adapted from The Guide to Investigation of Mouse Pregnancy, Croy, A. 2014 [3].

**Figure 3 cells-12-00620-f003:**
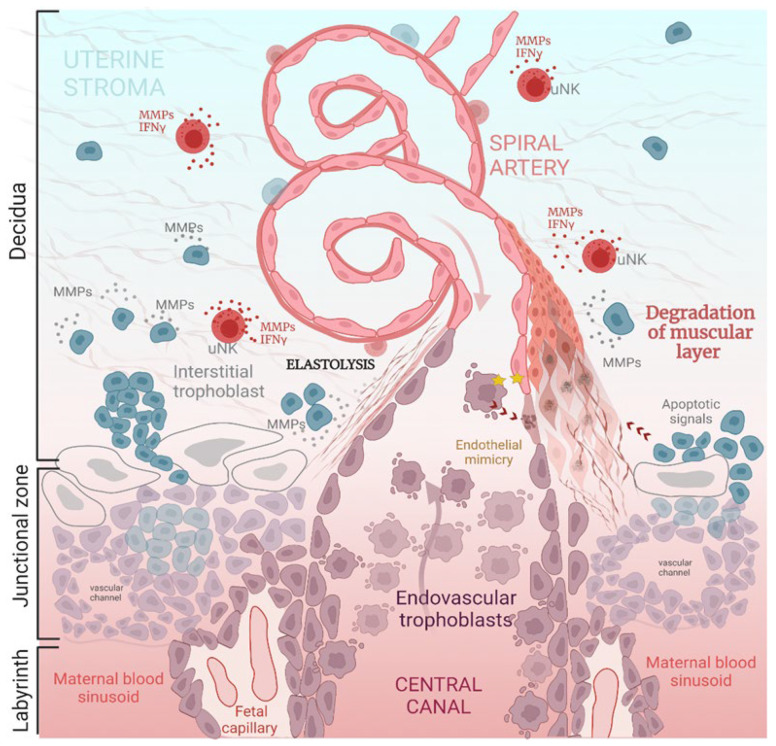
Schematic representation of the main mechanisms of spiral artery remodeling (SAR) in mice. Two main pathways of SAR are represented: the interstitial invasion of trophoblasts into the maternal decidua and the endovascular invasion of trophoblasts into the spiral arteries. The spiral arteries are partially invaded and then remodeled in the central canal by endovascular trophoblasts, which are called SpA-TGCs in mice (extravillous trophoblasts (EVTs) in humans). In addition, they infiltrate and replace endothelial cells via several mechanisms: endothelial mimicry, the induction of apoptosis, and the secretion of MMPs. They also induce the degradation of the vascular muscular layer and thus transform the spiral arteries into a central trophoblast-lined canal that carries maternal blood to the placental labyrinth. The maternal decidua is remodeled by interstitially invasive GlyT cells that originate from spongiotrophoblast (SpT) cells in the junctional zone. In addition, in mice, the junctional zone contains a layer of P-TGC cells that directly overlie the decidua but also the SpT and GlyT cells that surround the vascular channels and the central canal. Indeed, the invasive GlyTs secrete several MMPs and thus modify the composition of the uterine extracellular matrix. These GlyTs also induce the degradation of the arterial basal membrane (elastolysis) and SMCs. In addition, the decidual layer contains uNK cells, which participate in the SAR by secreting INFγ and MMPs. GlyT: glycogen trophoblast cell; uNK: uterine natural killer; SpA-TGC: spiral-arterial-associated trophoblastic giant cell; SpT: spongiotrophoblast; P-TGC: parietal trophoblast giant cell; EC: endothelial cell; SMC: smooth muscle cell; ECM: extracellular matrix; MMPs: matrix metalloproteinases; INF-γ: interferon gamma.

**Figure 4 cells-12-00620-f004:**
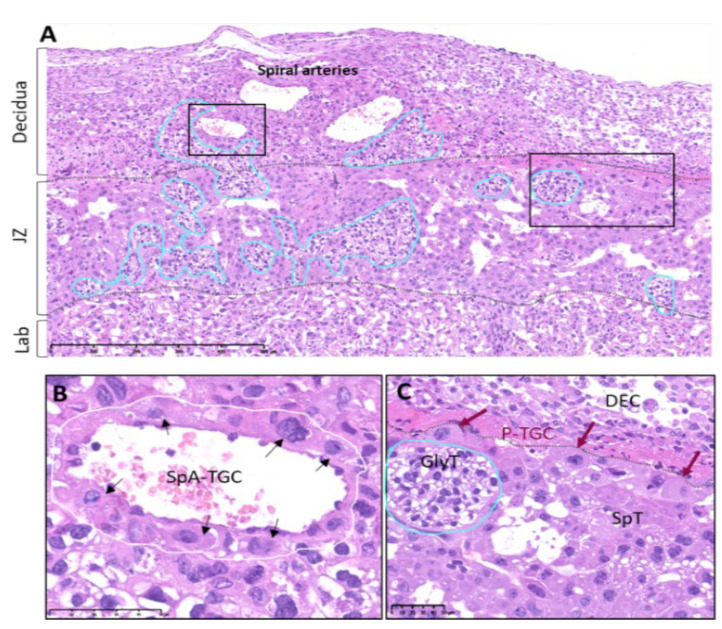
Histological images of mature murine placenta (E14.5) showing interstitial and endovascular remodeling pathways. (**A**) The central cross section shows the three major placental layers. Each is delineated by dotted lines. In addition, the group of GlyT cells migrates from the JZ (turquoise lines). The invading GlyTs remodel the uterine interstitium and reside in the perivascular spaces of the central canals. (**B**) A higher-magnification image shows the distal segments of the remodeled spiral arteries, which are lined by SpA-TGCs. (**C**) Higher magnification of the JZ highlights the presence of GlyTs, mixed with SpTs. Moreover, the P-TGCs are giant cells with enlarged hyperchromatic nuclei and abundant eosinophilic cytoplasm. They are in direct contact with the maternal decidua and anchor fetal-derived structures to the uterus.

## Data Availability

Not applicable.

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
