# Peer review of "Estrogen Actions in Placental Vascular Morphogenesis and Spiral Artery Remodeling: A Comparative View between Humans and Mice"

_cells, 2023, doi:10.3390/cells12040620_

Round 1

Reviewer 1 Report

This review seems to be in two sections: The first section comparing the placentation in murine and human pregnancy with a focus on spiral artery remodelling, which seems well described with some nice figures, and the last section, which describes the effects of estrogen on multiple targets in humans and a few animal models, with a focus on SAR haemodynamics. At the very end a murine knock-out model is described.

The purpose of the review is not clear, and reading the title and abstract first, there should be more focus on describing the Estrogenic actions, maybe with figures and more elaborate paragraphs. The manuscript now has less than four pages at the end discussing estrogen (out of 20 pages).

The manuscript needs English revision as there are some errors in sentences and words throughout the paper.

Some specific comments:

The title should be rewritten so it is clear that there is a focus on the murine model and SAR, and the similarities to human placenta.

P2.

The sections on mouse and human placenta are very different. It is very different what is described and at what part of the pregnancy (mouse at term, human through placentation). Please rewrite so the sections are comparable. Maybe you can describe the similar components first and then what differs. 

P4.

Fig 2: The bottom left picture of villous type placenta is not very illustrative - please depict fetal arteries and veins and the tissue surrounding them making up the villi.

P5.

The last paragraph: This statement is too broad. there are critical issues in using murine models in reproductive toxicology. Please be more specific or remove the sentence.

P18. The section on murine model for estrogen receptors does not fit this review. The model should be described in a paper on model-design and discussed properly.

Author Response

Reviewer 1: This review seems to be in two sections: The first section comparing the placentation in murine and human pregnancy with a focus on spiral artery remodelling, which seems well described with some nice figures, and the last section, which describes the effects of estrogen on multiple targets in humans and a few animal models, with a focus on SAR haemodynamics. At the very end a murine knock-out model is described.

The purpose of the review is not clear, and reading the title and abstract first, there should be more focus on describing the Estrogenic actions, maybe with figures and more elaborate paragraphs. The manuscript now has less than four pages at the end discussing estrogen (out of 20 pages).

We thank the reviewer for his/her comments that help us to improve this review. We have summarized the comparative view of the human and mouse placenta and the comparative anatomy to the fetal and maternal circulation, into 3 pages in order to emphasize the last part of the review where we describe in more details all the multiple actions of estrogens on the  main vasculogenic events during pregnancy. We hope that this review is now more focused.

The manuscript needs English revision as there are some errors in sentences and words throughout the paper.

We have tried our best to improve the English editing.

Some specific comments:

The title should be rewritten so it is clear that there is a focus on the murine model and SAR, and the similarities to human placenta.

The title has been changed and the first part has been conserved, stating only “Estrogen actions in the placental vascular morphogenesis and spiral arterial remodeling” because we tried to emphasize the effects of estrogens.

P2. The sections on mouse and human placenta are very different. It is very different what is described and at what part of the pregnancy (mouse at term, human through placentation). Please rewrite so the sections are comparable. Maybe you can describe the similar components first and then what differs. 

We have shortened this part and rewrote these sections to better compare the morphology and the vascular system of the human and mouse placenta. This is now summarized into 2 pages, and 2 main figures.

P4. Fig 2: The bottom left picture of villous type placenta is not very illustrative - please depict fetal arteries and veins and the tissue surrounding them making up the villi.

This figure has been modified with captions referring to fetal capillaries and maternal blood. The Figure 2 has also been combined with the original Figure 1.

P5. The last paragraph: This statement is too broad. there are critical issues in using murine models in reproductive toxicology. Please be more specific or remove the sentence.

All this part has been modified and synthesized.

P18. The section on murine model for estrogen receptors does not fit this review. The model should be described in a paper on model-design and discussed properly.

We agree with the reviewer that this paragraph was too long and was not fitting well. This paragraph has been completely removed and this murine model has been discussed properly, in 2 different sections, first on page 14 reporting the  estrogen impact on uterine arterial remodeling “However, a more recent study assessing ex vivo uterine arterial dilation in response to applied pressure did not demonstrate any alteration of pregnancy-associated uterine arterial remodeling in C451A-ERa mice deficient in membrane ERα signaling at E9.5, suggesting normal maternal blood supply to the downstream spiral arteries [131], [132]. Thus, this study strongly indicates that membrane-initiated ERa signaling is not required for the arteriolar remodeling in uterine arteries in response to the physiological increase in blood flow during pregnancy. “

and secondly on page 15, reporting the estrogen membrane action on specific spiral arterial remodeling.  “Indeed, C451A-ERa mice show several abnormalities in gestation and parturition, leading to final total neonatal lethality of the offspring [132]. We demonstrated that the partial embryonic death appearing at about E9.5, results from both the impaired expansion of Tpbpa-positive spiral artery-associated trophoblast giant cells (SpA-TGCs) in the utero-placental unit, and the imbalance in the expression of angiogenic factors and Notch family members, together controlling the remodeling of the maternal spiral arteries.”

Reviewer 2 Report

The manuscript reviews the recent data on placental vascular morphogenesis and spiral arterial remodeling. The focus in this manuscript is on comparative view between humans and mice. In addition, estrogen actions in the placental vascularization are briefly described. The manuscript explains understandably with supporting figures the complex placental vascularization both in mouse and human. Mouse is a widely used model in studying human reproduction. To understand human diseases in mouse models, it is necessary to understand differences and similarities between species. The comprehensive comparison of mouse and human placenta based on current knowledge is the main strength of this manuscript.

The comparison between mouse and human placental vascular morphogenesis and spiral arterial remodeling is well and comprehensively described in this manuscript. This is a relevant topic and useful for other researchers, because mouse is widely used in studying human reproduction, and there are no similar review articles published before. However, unlike the title and abstract explain, the estrogen action in this process is not the focus of this review. Especially, the data on the role of estrogen in mouse placental development is based on one article only, published by the authors. Thus, the title and abstract should be rewritten accordingly.

 Although, the comparison between species was adequate, sometimes it was difficult to follow if the text refers to mouse or human. This should be clearly indicated either in the titles or in the text. The authors should go through the whole text with this aspect in their minds.

Specific comments:

 1.       Referring to the previous chapter describing the inaccuracy in describing the species: Chapter V, the title is “Overview of the hemochorial placentation in mouse”, subtitle C:”spiral arterial remodeling (SAR)”, The second paragraph defines the endometrial changes during menstrual shedding, which does not exist in mouse. This has not been mentioned.

2.       Similarly, chapter XI, clearly presents functions in human (e.g. placental E2 production, menstrual cycle), but it has not been particularized in the titles or text.

3.       Please, check the whole manuscript and make sure that it is always clear for the reader, which species the text concerns.

4.       Page 9, the fourth line in the paragraph B: Should it refer to figure 4 instead of figure 3?

5.        Title IX “Steroid hormonal control of reproductive actions during pregnanacy” and subtitle A “Overview of the estrogen actions during pregnancy”. Both titles are too broad. Please, change them to more focused ones.

Author Response

 Reviewer 2:

The manuscript reviews the recent data on placental vascular morphogenesis and spiral arterial remodeling. The focus in this manuscript is on comparative view between humans and mice. In addition, estrogen actions in the placental vascularization are briefly described. The manuscript explains understandably with supporting figures the complex placental vascularization both in mouse and human. Mouse is a widely used model in studying human reproduction. To understand human diseases in mouse models, it is necessary to understand differences and similarities between species. The comprehensive comparison of mouse and human placenta based on current knowledge is the main strength of this manuscript.

The comparison between mouse and human placental vascular morphogenesis and spiral arterial remodeling is well and comprehensively described in this manuscript. This is a relevant topic and useful for other researchers, because mouse is widely used in studying human reproduction, and there are no similar review articles published before. However, unlike the title and abstract explain, the estrogen action in this process is not the focus of this review. Especially, the data on the role of estrogen in mouse placental development is based on one article only, published by the authors. Thus, the title and abstract should be rewritten accordingly.

 Although, the comparison between species was adequate, sometimes it was difficult to follow if the text refers to mouse or human. This should be clearly indicated either in the titles or in the text. The authors should go through the whole text with this aspect in their minds.

We have completely remodeled this review trying to be more specific in the title of the different chapters in order to organize the main actions of estrogens in the major placental vascular events. We hope this is now more explicit for the reader.

Specific comments:

  1. Referring to the previous chapter describing the inaccuracy in describing the species: Chapter V, the title is “Overview of the hemochorial placentation in mouse”, subtitle C:”spiral arterial remodeling (SAR)”, The second paragraph defines the endometrial changes during menstrual shedding, which does not exist in mouse. This has not been mentioned.

This part has been removed.

  1. Similarly, chapter XI, clearly presents functions in human (e.g. placental E2 production, menstrual cycle), but it has not been particularized in the titles or text.

It is not clear what is the reviewer asking for. Sorry if we missed the point.

  1. Please, check the whole manuscript and make sure that it is always clear for the reader, which species the text concerns. We have tried our best to include this information with the citation mentioned.
  2. Page 9, the fourth line in the paragraph B: Should it refer to figure 4 instead of figure 3?

This figure has been deleted, as this part referred only to mice and no similar description was made for humans. The first 10 pages has been highly synthesized to shorten the introduction and focused this review on the estrogen impact on placental vascular morphogenesis and remodeling.

  1. Title IX “Steroid hormonal control of reproductive actions during pregnanacy” and subtitle A “Overview of the estrogen actions during pregnancy”. Both titles are too broad. Please, change them to more focused ones.

The entire review has been reorganized. The titles of the sections have therefore been changed.

Reviewer 3 Report

This review provides a good overview of the role of estrogen in the placental vascular morphogenesis and spiral arterial remodeling, and the author chose to compare it in mice and humans. The article covers a very comprehensive, detailed induction and classification, and analyzes and refines the articles published in recent years. However, I am concerned about the following questions.

1. Keyword selection. This review focuses on the role of estrogen. It's better to replace the estrogen receptor with estrogen.

2. The background of Figure3 needs to be adjusted. The color is too dark and it is difficult to see the font in the figure clearly. All cell types or factors present in the picture need to be clearly marked, such as endothelial cells and red blood cells. Similarly, the background of Figure2 is very deep and the font is not legible. If it needs to be highlighted, zoom in or zoom out by the same factor.

3. The author cites previous studies well, but lacks of analyzing the development status and application prospects of this field from his own perspective. It is better to elaborate the future research prospects for more people to generate research interest.

4. The author addresses several unresolved scientific questions in the passage. For example, “Nevertheless, the initiation signals and the regulatory mechanisms of these cellular transformations still remain unknown” and “Nevertheless, the precise mechanisms of their regulations like their murine analogues still remain unknown”. The author should discuss it and give their own opinions based on their own research. The author can explore different aspects of the answer based on important research, including what he agrees with and what he disagrees with.

5. The author can improve the coherence of the review by focusing on one main line throughout. The author's first description of placental physiology in mice and humans is too much, this review needs to emphasize the regulatory role of estrogen. Highlighting the main points will make it easier for the reader to accept the points made by the author.

6. The author can discuss the advantages and disadvantages of his own research in the proposed subtitle based on his own research experience, compare it with others' research, and what innovations need to be made to contribute to future research.

Author Response

This review provides a good overview of the role of estrogen in the placental vascular morphogenesis and spiral arterial remodeling, and the author chose to compare it in mice and humans. The article covers a very comprehensive, detailed induction and classification, and analyzes and refines the articles published in recent years. However, I am concerned about the following questions.

  1. Keyword selection. This review focuses on the role of estrogen. It's better to replace the estrogen receptor with estrogen.

We are in full agreement on this point. The selection of keywords has been modified based on the reviewer's comment.  

  1. The background of Figure3 needs to be adjusted. The color is too dark and it is difficult to see the font in the figure clearly. All cell types or factors present in the picture need to be clearly marked, such as endothelial cells and red blood cells. Similarly, the background of Figure2 is very deep and the font is not legible. If it needs to be highlighted, zoom in or zoom out by the same factor.

Figure 2 has been modified. The font has been changed and the colors highlighted. It has also been combined with original Figure 1. We hope that this new figure helps to better explicit the comparison of the morphology of mouse and human placentas.

  1. The author cites previous studies well, but lacks of analyzing the development status and application prospects of this field from his own perspective. It is better to elaborate the future research prospects for more people to generate research interest.

We have included some comments on future elaborate works:

  1. On page 8: “With the development of all the new omics technologies such as single-cell and spatial transcriptomics, a direct correlation of single-cell RNA profiles and the exact morphological localization of a cell thus remains to be established.”
  2. On page 14, paragraph 2, “This is probably because the contribution of NO to these changes appears to be modest, as uterine arterial remodeling was impaired but not absent in pregnant Nos3 knock-out mice”
  • On page 14, paragraph 3, However, many works using ERβ antibodies need to be re-evaluated since Andersson et al, [134] questioned the validation of the former antibodies against ERβ.
  1. On page 15, paragraph 4: “ It should be noted, that generally, the role of estrogen receptors in murine spiral arterial remodeling has received limited attention, given that all mouse models invalidated for ERs isoforms (both ERα and ERβ) or its sub-functions (ERα-AF1° and ERα-AF2°) are infertile [70]. Thus, C451A-ERa mouse [131], mutated for the palmitoylation site of ERα that specifically abrogated ERα membrane expression, appears as a first mouse model to study the function of maternal ERα in trophoblast development and pregnancy during early placentogenesis, in vivo.”
  2. We have also extended the conclusion: “In conclusion, estrogen signaling in utero-placental vascular morphogenesis remains a fascinating but understudied area of research. Moreover, it is a critical process whose alteration may jeopardize the well-being of the mother and the development of the fetus. The mouse is recognized as an ideal animal model to study the physiology and pathophysiology of pregnancy in mammals because of all its reproductive advantages, the relative simplicity of genetic manipulation, ……Nevertheless, although some pathological mechanisms may not be directly applicable to humans, the various mouse models of pregnancy may shed new light on the fundamental mechanisms of maternal adaptation regulated by estrogen during gestation. This could lead to the identification of therapeutic targets or even preventive strategies for pregnancy disorders and consequently, could reduce perinatal mortality and morbidity. “

  1. The author addresses several unresolved scientific questions in the passage. For example, “Nevertheless, the initiation signals and the regulatory mechanisms of these cellular transformations still remain unknown” and “Nevertheless, the precise mechanisms of their regulations like their murine analogues still remain unknown”. The author should discuss it and give their own opinions based on their own research. The author can explore different aspects of the answer based on important research, including what he agrees with and what he disagrees with.

See comment N° 2 and comment N°3 above.

  1. The author can improve the coherence of the review by focusing on one main line throughout. The author's first description of placental physiology in mice and humans is too much, this review needs to emphasize the regulatory role of estrogen. Highlighting the main points will make it easier for the reader to accept the points made by the author.

We have completely redesigned this review by trying to synthesize the first part on placental physiology in mice and humans in 3 pages. We then emphasized the regulatory roles of estrogens on the major vascular events during pregnancy. We hope that it is now more organized and easier for the reader.

  1. The author can discuss the advantages and disadvantages of his own research in the proposed subtitle based on his own research experience, compare it with others' research, and what innovations need to be made to contribute to future research.

We have added our own research on page 13: “Thus, this study strongly indicates that membrane-initiated ERa signaling is not required for the arteriolar remodeling in uterine arteries in response to the physiological increase in blood flow during pregnancy.

 And on page 15:, “ It should be noted, that generally, the role of estrogen receptors in murine spiral arterial remodeling has received limited attention, given that all mouse models invalidated for ERs isoforms (both ERα and ERβ) or its sub-functions (ERα-AF1° and ERα-AF2°) are infertile [70]. Thus, C451A-ERa mouse [131], mutated for the palmitoylation site of ERα that specifically abrogated ERα membrane expression, appears as a first mouse model to study the function of maternal ERα in trophoblast development and pregnancy during early placentogenesis, in vivo

Round 2

Reviewer 1 Report

The title does not cover the two species that you focus on, I suggest adding somewhere “in humans and murine models”

The paper is very broad and lacks focus. There is a nice section comparing murine and human placentas, a nice section on SAR, and at the end a lengthy and complicated section on the effects of estrogen. This section would benefit from a figure depicting the different pathways of estrogen effects on the implantation, placenta and fetus, alternatively a table with the effects of abnormal estrogen/ER and the effects of raised vs normal vs lowered estrogen concentration, with species differences –maybe table 2 could be extended? I suggest focusing the paper further to deliver your message clearly. Some English language editing is still required.

P2 last paragraph: remove ”As” and the comma after “humans”.

Figure 2 text: Reference at the end should be in the reference list with a reference number.

P4. The mouse placenta is highly efficient in the counter-current blood flow exchange. What is the advantage of the less efficient multivillous flow exchange found in humans?

P9 IV They exert pleiotropic.

P9 3rd section: Although the changes… during pregnancy. Rewrite the sentence to aid clarity.

P9-10 “big apes” –change to “higher primates”.

The sections on effects of estrogen could somewhere mention the feedback and feedforward mechanisms in the pregnant woman.

P13 1st paragraph: “… intrauterine growth retardation or preecclamsia, it occurs in all mammalian species studied”. Preecclampsia is not as far as I know found in other species than human, and I am not sure how frequent preterm delivery and IUGR is. Please rephrase the statement.

You state multiple times in the section and throughout the paper that the increase of estrogen correlates with something, but correlation does not in itself suggest causation. Please rephrase to not directly suggest that correlation is proof of causation.

3rd paragraph: “Therefore E2 fascilitates…” you have not presented the data to support this statement.

P16 last paragraph before V: “Taken together, these data suggest” Please remove this statement, as it contradicts the statement directly before it.

In the conclusion there is a glorification of the murine model that is inappropriate. The 3rd sentence stating “the mouse is recognized as an ideal animal model… considerations” should be removed and the conclusion should state objectively the possibilities available to study the implantation process in humans.

Author Response

Comments and Suggestions for Authors

The title does not cover the two species that you focus on, I suggest adding somewhere “in humans and murine models”

We added on the title: comparative view between humans and mice

The paper is very broad and lacks focus. There is a nice section comparing murine and human placentas, a nice section on SAR, and at the end a lengthy and complicated section on the effects of estrogen. This section would benefit from a figure depicting the different pathways of estrogen effects on the implantation, placenta and fetus, alternatively a table with the effects of abnormal estrogen/ER and the effects of raised vs normal vs lowered estrogen concentration, with species differences –maybe table 2 could be extended? I suggest focusing the paper further to deliver your message clearly. Some English language editing is still required.

We added a graphical abstract that summarizes the different effects of estrogens over the course of pregnancy. We also reviewed the Table 2 to make it more straightforward. However, there is a lot of effects of estrogens that are difficult to summarize easily. We tried to clarify and cut some points on estrogen effects at the end of the manuscript. We hope that our changes are now helping to deliver a clear message.

Regarding the English editing, we have tried our best to improve it. In particular, we remove some expressions that were largely repeated such as “ it appears to”, It should be noted… and we rephrased some sentences. If you think there is too many mistakes, we can ask to review the complete manuscript by qualified English speaking editing services.

P2 last paragraph: remove ”As” and the comma after “humans”.

This has been done.

Figure 2 text: Reference at the end should be in the reference list with a reference number.

This reference is number 3, and has been mentioned.

P4. The mouse placenta is highly efficient in the counter-current blood flow exchange. What is the advantage of the less efficient multivillous flow exchange found in humans?

We added some comments to answer that question:“ It is believed that the villous type placenta is less efficient but can compensate by increasing its mass. Moreover, the human villous placenta imposes less metabolic demand on the mother than the labyrinthine type. This enables a low daily fetal growth rate and a longer gestation for humans, as they tend to produce larger offspring. Therefore, in humans, this exchange between villous trophoblast and maternal blood represents an evolutionary compromise that helps to maintain the balance between maternal and fetal demands on longer gestations without depleting maternal resources [8]. 

P9 IV They exert pleiotropic. Has been changed

P9 3rd section: Although the changes… during pregnancy. Rewrite the sentence to aid clarity.

This sentence has been rewritten and some precision was added: “Endocrine function differs in human and mouse placentas [50]. In mice, the corpus luteum has to produce progesterone throughout gestation. The second difference is that genes encoding enzymes involved in steroidogenesis are not expressed in the mouse placenta during the second half of gestation, whereas they are expressed in humans late in gestation. Accordingly, although ovaries are generally the main producer of hormones,  the human placenta becomes the main organ producing estrogens during pregnancy [50], [51]

P9-10 “big apes” –change to “higher primates”. Change was done

The sections on effects of estrogen could somewhere mention the feedback and feedforward mechanisms in the pregnant woman.

We have not included this point since we did not understand the question you want to address.

P13 1st paragraph: “… intrauterine growth retardation or preecclamsia, it occurs in all mammalian species studied”. Preecclampsia is not as far as I know found in other species than human, and I am not sure how frequent preterm delivery and IUGR is. Please rephrase the statement.

You are completely right. Thanks for mentioning it. This was a mistake, because this difference between mice and humans was mentioned in the conclusion.  We have removed the following sentence in this paragraph: “It occurs in all mammalian species studied.” And we have modified the sentence to make it clear “or even pre-eclampsia that is only observed in humans.”

You state multiple times in the section and throughout the paper that the increase of estrogen correlates with something, but correlation does not in itself suggest causation. Please rephrase to not directly suggest that correlation is proof of causation.

We totally agree, but in humans it is difficult to avoid correlation. So, we took into account your comment and we were more careful in the way to report these data and we have changed some of these sentences:

In particular on page 13: we removed the causation. “In most mammalian pregnancies, the progressive increase in E2 levels coincides with the several-fold increase in the utero-placental blood flow, suggesting its central role.”

Also  “The increase in weight paralleled the increase in plasma E2 levels, suggesting  its pivotal role over there” was changed to: “The increase in weight paralleled the increase in plasma E2 levels, suggesting  a possible role of estrogens on this uterine arterial adaptation”

On page 15: “Interestingly, premature elevation of its circulating concentration above the upper range during the first trimester alters trophoblast survival through its pro-apoptotic and anti-proliferative influence, suggesting a possible interplay between elevated serum E2 and apoptosis in the first trimester of pregnancy [135] . We also remodeled this paragraph to make it easy to read.

3rd paragraph: “Therefore E2 fascilitates…” you have not presented the data to support this statement.

We have preferred to remove this paragraph that was summarizing all the data presented above. But this was indeed unclear.

P16 last paragraph before V: “Taken together, these data suggest” Please remove this statement, as it contradicts the statement directly before it.

This paragraph has been removed

In the conclusion there is a glorification of the murine model that is inappropriate. The 3rd sentence stating “the mouse is recognized as an ideal animal model… considerations” should be removed and the conclusion should state objectively the possibilities available to study the implantation process in humans.

“Ideal model” was probably inappropriate. So we changed that sentence into: “ Although mouse and human placental structures are different, mouse models have been instrumental for studying the physiopathology of mammalian pregnancy, because of all their reproductive advantages, the similarities between mice and placental development (Table 1), the relative simplicity of genetic manipulation, and ethical and cost considerations”

See the manuscript enclosed with the graphical abstract included.

Reviewer 2 Report

This manuscript can now be accepted.

Author Response

Thanks for your acceptance.

You will find below the last versison where we have revised the section B and C of the last section.

Thanks again

Round 3

Reviewer 1 Report

The graphical abstract is excellent, and a great addition to the paper!

Figure 3 is also great!

P12 IV. In the non-pregnant womaen estrogen fluctuates during the menstrual cycle and signals the  anterior pituitary in both a feedback and feed-forward loop to stimulate LH and FSH and cause ovulation. I think this should be mentioned in this section as these mechanisms start the pregnancy, and also what causes the suppression of the ovarian secretion of estrogen by the placental estrogen.

Section 4. B and C. are still very heavy with information. What are the key messages of these sections? Maybe you can select some key points and divide the sections further.

Author Response

The graphical abstract is excellent, and a great addition to the paper!

Figure 3 is also great!

Thanks for your enthusiasm.

P12 IV. In the non-pregnant women estrogen fluctuates during the menstrual cycle and signals the  anterior pituitary in both a feedback and feed-forward loop to stimulate LH and FSH and cause ovulation. I think this should be mentioned in this section as these mechanisms start the pregnancy, and also what causes the suppression of the ovarian secretion of estrogen by the placental estrogen.

This comment has been added on Page 12: “In the non-pregnant women, estrogen fluctuates during the menstrual cycle and signals the anterior pituitary in both a feedback and feed-forward loop to stimulate LH and FSH and cause ovulation.   

Section 4. B and C. are still very heavy with information. What are the key messages of these sections? Maybe you can select some key points and divide the sections further.

These parts have been re-organized and completely re-written with subtitles in the section.

We are grateful to the reviewer who helped us to improve this review.

All the best

Francoise LENFANT